# Microstructure and Mechanical Properties of Rolled (TiC + Ti1400)/TC4 Composites

**DOI:** 10.3390/ma18010051

**Published:** 2024-12-26

**Authors:** Bowen Li, Shanna Xu, Ni He, Guodong Sun, Mingyang Li, Longlong Dong, Mingjia Li

**Affiliations:** 1School of Materials Science and Engineering, Xi’an Shiyou University, Xi’an 710065, China; libw1110@163.com; 2Xi’an Rare Metal Materials Institute Co., Ltd., Xi’an 710016, China; hn1239210639@163.com (N.H.); guodongsun126@126.com (G.S.); myli.ustb@gmail.com (M.L.); 3Advanced Materials Research Central, Northwest Institute for Nonferrous Metal Research, Xi’an 710016, China; donglong1027@163.com

**Keywords:** titanium matrix composites, reinforcements, microstructures

## Abstract

One of the long-standing challenges in the field of titanium matrix composites is achieving the synergistic optimization of high strength and excellent ductility. When pursuing high strength characteristics in materials, it is often difficult to consider their ductility. Therefore, this study prepared a Ti1400 alloy and in situ synthesized TiC-reinforced (TiC + Ti1400)/TC4 composites using low-energy ball milling and spark plasma sintering technology, followed by hot rolling, to obtain titanium matrix composites with excellent mechanical properties. The Ti1400 alloy bonded well with the matrix, forming uniformly distributed Ti1400 regions within the matrix, and TiC particles were discontinuously distributed around the TiC-lean regions, forming a three-dimensional network structure. The (TiC + Ti1400)/TC4 composites effectively enhanced their yield strength to 1524 MPa by using 3 wt.% of Ti1400 alloy while preserving an impressive elongation of 9%. When the Ti1400 alloy content reaches 20 wt.%, the overall mechanical properties of the composites decrease. A small amount of Ti1400 does not reduce the strength of the composite. On the contrary, it can undergo stress-induced phase transformation during plastic deformation, thereby coordinating deformation, which not only provides higher strain hardening and increases tensile strength but also benefits ductility.

## 1. Introduction

Due to their excellent mechanical properties, excellent corrosion resistance, and good biocompatibility, titanium and its alloys have been widely applied and extensively studied in key fields such as aerospace, medical devices, chemical engineering, and the military industry [1,2,3,4,5,6]. By introducing ceramic phases or whiskers as reinforcements into the titanium alloy, it is possible to construct titanium matrix composites (TMCs) with high specific strength, excellent heat resistance, and lower cost [7,8,9,10,11,12]. Ti-6Al-4V (TC4) is a typical commercialized mature titanium alloy, which has higher specific strength, lower density, and better overall mechanical properties compared to commercial pure titanium (CP-Ti); hence, a multitude of studies opt for TC4 as the matrix material [13,14,15,16,17,18]. Achieving good strength–ductility matching in TMCs hinges on the selection of appropriate reinforcement.

In the design of TMCs, the reinforcement plays a central role and must possess exceptional mechanical properties, including high strength, a high modulus, and excellent load transfer ability [19,20,21]. To minimize the deformation differences between the reinforcement and the matrix during preparation and use, the reinforcement must have a linear thermal expansion coefficient comparable to that of the matrix. Additionally, the reinforcement must demonstrate excellent compatibility with the matrix material, thermodynamic stability, and the ability to form a strong interfacial bond, which are the basic criteria for evaluating its suitability as a reinforcement. By adjusting these properties, the reinforcement can better withstand temperature fluctuations and mechanical stresses, ensuring that the composite structure is more stable and reliable [22,23,24,25,26]. Common reinforcements include carbides and borides (such as TiB and TiC), metal oxides (like Al_2_O_3_), and rare earth oxides (such as Y_2_O_3_ and La_2_O_3_), which are widely studied and applied due to their unique physicochemical properties [27,28,29,30,31,32,33]. TiC as a reinforcement is an ideal choice due to its similar physical properties to the titanium alloy, such as density, Poisson’s ratio, and thermal expansion coefficient, as well as its stable chemical properties. These characteristics help reduce deformation differences between the reinforcement and the matrix, thereby enhancing the overall performance of the composites [34,35,36,37,38,39]. Huang et al. have shown that introducing (Ti_5_Si_3_ + TiC_x_) (x = 0.67) particles into the TC4 matrix can significantly improve the material’s yield strength and ultimate compressive strength [40]. Liu et al. utilized spark plasma sintering (SPS) technology combined with subsequent heat treatment to prepare graphene nanoparticles (GNPs)/titanium composites, a method that enhances strength without sacrificing the material’s total elongation [41]. The application of TiC as a reinforcement in TMCs, by optimizing its interfacial bonding with the matrix and controlling its distribution, can effectively improve the material’s comprehensive performance. At the same time, the use of advanced preparation techniques such as SPS can further enhance the performance of composites, providing new directions for the development of TMCs.

Despite the superior mechanical properties that TiC possesses as a reinforcement in TMCs, it remains a challenging material due to its inherent hardness and brittleness [42]. This brittleness is a primary source of cracks within the composites, hindering the material’s ability to achieve significant elongation or stretchability. Consequently, enhancing the ductility of TMCs reinforced with TiC remains a significant challenge in materials science.

Researchers have successfully prepared multi-principal element alloy composites with high strength and toughness by using the AlNbTiVZr high-entropy alloy as the reinforcement and mixing it with the TA15 alloy. This strategy has been shown to increase the material’s yield strength to about 1300 MPa while maintaining an elongation of around 8%, demonstrating excellent performance [43]. This indicates that selecting an alloy with a similar crystal structure to the matrix and good bonding without forming intermetallic compounds as the reinforcement is an effective method to improve both strength and plasticity, achieving a strength–ductility match in composites [44,45]. The Ti1400 (Ti-Al-V-Cr-Mo) alloy, as an age-hardened high-strength titanium alloy, is expected to achieve a good strength–ductility match in (TiC + Ti1400)/TC4 composites due to its α-phase and β-phase crystal structures being consistent with those of the TC4 alloy, without forming intermetallic compounds.

Previously, He et al. prepared (TiC + Ti1400)/TC4 by spark plasma sintering, and the properties were tensile strength of 1178 MPa and elongation of 12.8% [46]. However, for TMCs, strength over 1500 MPa and elongation over 5% are still a challenge to achieve. Thermomechanical processing, such as hot rolling, is a common method used to modulate the microstructure and mechanical properties of titanium alloys and TMCs. The strengthening mechanisms it brings, such as grain refinement and dislocation strengthening, will be an effective means to achieve the extraordinary mechanical properties of TMCs. Therefore, in this paper, (TiC + Ti1400)/TC4 was prepared by the SPS method, and in order to avoid cracking due to the poor plasticity of TiC and the severe oxidation of the titanium alloy matrix during the hot rolling process, multiple passes were carried out below the temperature of the phase transition point of TC4 (1173 K). In this experiment, the rolled (TiC + Ti1400)/TC4 composites were characterized using optical microscopy (OM), tensile testing, scanning electron microscopy (SEM), Vickers hardness, and electron backscatter diffraction (EBSD), and the strengthening mechanisms of the composites were discussed.

## 2. Experimental Procedures

### 2.1. Materials Preparation

TC4 spherical powder (15~53 μm, 99.9% purity, Xi’an Sino-European Materials Co., Ltd., Xi’an, China), Ti1400 spherical powder (Xi’an Sino-European Materials Co., Ltd., Xi’an, China), and carbon nanoparticles (20 nm, purity ≥ 99.8%, Shaanxi Coal Chemical Group Co., Ltd., Xi’an, China) are original materials for the preparation of composites. Table 1 demonstrates the content of each element in the original powder. The manufacturer provided the chemical elements of the TC4 original powder. The chemical elements of Ti1400 powder were measured by the authors by inductively coupled plasma atomic emission spectrometry. In Table 1, “/” means that the element was not measured.

Figure 1 shows the microscopic morphology of the powders and the preparation process of (TiC + Ti1400)/TC4 composites. Figure 1a shows the morphology of CNPs, and CNPs are irregular particles of nanometer size. The microstructure of TC4 and Ti1400 original powders are both regular spherical particles (Figure 1b,c). From Figure 1(b_1_,c_2_), the Grain Statistics software (Nano Measurer 1.2) determined that the average particle diameters of TC4 and Ti1400 powders were 50.18 µm and 22.53 µm, respectively. In this series of composites, the added Ti1400 content was 0 wt.%, 3 wt.%, 20 wt.%, and 0.5 wt.%. CNPs were added to all samples. The samples were named TMC-0 (0.5 wt.% CNPs/TC4), TMC-3 (3 wt.% Ti1400 + 0.5 wt.% CNPs/TC4), and TMC-20 (20 wt.% Ti1400 + 0.5 wt.% CNPs/TC4). Figure 1d shows the process of low-energy ball milling by adding the powders into a ball milling tank, where the ball milling speed was 200 rpm and the ball-to-material ratio was 5:1. To ensure homogeneous mixing, half of the reinforcements were added in the beginning, and after mixing for 1 h, all the rest of the reinforcements were added, and ball milling was carried out for 6 h. After mixing, (TiC + Ti400)/TC4 composites were prepared by SPS at 1273 K for 5 min under 40 MPa, and the vacuum was kept at 10^−2^ Pa. The TiC particles could be synthesized according to the following reaction (1):Ti(s) + C(s) → TiC(s)(1)

The size of the sintered samples was Φ 50 × 12 mm. Cylindrical samples of 8 mm in thickness were cut from the sintered samples for rolling. Before rolling, the samples were kept warm in a muffle furnace at 1173 K for 20 min. The samples were rolled from 8 mm to 2 mm in 9 passes by a two-high mill with a model of Φ 200 × 300, and the deformation was 75%. The RD direction is the direction along which the sample moves as it passes through the roll, called the rolling direction, abbreviated as RD. ND is the direction normal to the face in contact with the roll, and the direction perpendicular to both RD and ND is TD.

### 2.2. Material Characterization

An X-ray diffractometer (XRD, Mini-Flex600, Rigaku Corporation, Akishima, Japan) was used to characterize the phase composition of the composites. The scanning speed was 10°/min with a step size of 0.01 µm in the 2θ range of 30°~90°. The samples were etched after grinding and polishing using an etching solution with the composition of (HF:HNO_3_:H_2_O = 2:3:100). The metallographic organization of the samples was observed using a optical microscope (Zeiss-AXIO, Carl Zeiss AG, Jena, Germany). The etched samples were observed using a cold field emission scanning electron microscope (JSM-7500F, Nippon Tsunen Industrial (Hong Kong) Co., Ltd., Hong Kong, China). The samples for tensile testing had been cut in the rolling direction. The tensile samples were cut by wire-cutting into dog-bone-shaped tensile specimens with dimensions of 70 mm × 11 mm × 2 mm. Tensile tests were carried out using a UTM5105X electronic universal testing machine (manufactured by Shenzhen Sansi Zongheng Technology Co., Ltd., Shenzhen, China) with a tensile speed of 1 mm/min. The tensile tests were repeated three times. The mean and standard deviation were calculated and plotted. The metallographic samples were utilized for the Vickers hardness test using an HVS1000 digital automatic turret microhardness tester (manufactured by Laizhou Lelote Experimental Instrument Co., Ltd., Laizhou, China), with a force of 0.1 Kg and a retention time of 15 s. The Vickers hardness test was repeated for each area, and the mean and standard deviation were calculated. The samples for EBSD were mechanically ground to 0.5 µm and then electropolished in a solution of 6% HClO_4_/34% CH_3_(CH_2_)_3_OH/60% CH_3_OH at 35 V for 25 s. EBSD was performed on a scanning electron microscope (SEM, SU3500, Hitachi, Tokyo, Japan). The electron beam was moved in steps of 0.5 µm over an area of 200 × 200 µm.

## 3. Results and Discussion

### 3.1. Microstructure of (TiC + Ti1400)/TC4 Composites

Figure 2 shows the XRD spectra of (TiC + Ti1400)/TC4 composites. In the plots of TMC-0, TMC-3, and TMC-20, there is no significant change in the peaks of the α-phase and TiC. The (110) plane of the β-phase appears in TMC-3, which corresponds to an angle of 2θ = 39.7°. Figure 2b is a magnified view of the yellow dashed box in Figure 2a, and the diffraction peaks of the β-phase increase with the increase in Ti1400. The reason for the absence of β-phase diffraction peaks in TMC-0 may be due to the fact that the volume fraction of the β-phase in TC4 is already small, and it is difficult to characterize the β-phase by XRD because of the grain fragmentation after the hot rolling process. Since the hot rolling temperature is 1173 K, which is higher than the phase transition point of Ti1400, and the specimen is thin after rolling, even if air cooling, the cooling speed is also very fast, and more β-phase is obtained, which can be characterized by XRD. With the increase in Ti1400 content, the diffraction peaks of the β-phase are enhanced.

Figure 3 presents the OM images of TMC-0 and TMC-20. The gray area in Figure 3a represents the TC4 matrix, where the α-phase and β-phase are mixed. The magnified image in Figure 3b shows that the α-phases and β-phases are fragmented after the hot rolling. The hot rolling effectively promotes the fragmentation and recrystallization of the α-phases and β-phases, resulting in finer grain sizes and a uniform microstructure. Meanwhile, the uniform distribution of TiC particles in the matrix after hot rolling indicates that hot rolling can enhance the interfacial bonding between the particles and the matrix. In Figure 3c, with the increase in Ti1400 content, many white areas appear in the matrix, which are the Ti1400 regions. Figure 3d is a high-magnification image of Figure 3c, where the Ti1400 reinforcement and TC4 matrix are well bonded without obvious voids or cracks, which indicates that the preparation process of the composites can achieve good interfacial bonding.

Figure 4 displays the microstructures of TMC-0, TMC-3, and TMC-20. From Figure 4a, it is evident that in the absence of Ti1400 alloy addition, the TC4 matrix primarily consists of α-phases and β-phases; part of the β-phase after rolling is elongated along the RD direction, and part of it is elongated along the TD direction, while the α-phase is fragmented, forming a short bar-shaped organization. The TiC particles distribute almost parallel to the rolling direction, creating a discontinuous network structure. As the content of the Ti1400 alloy increases, more white regions appear in the matrix. These regions are composed of the β-phase from the Ti1400 alloy and are uniformly distributed with the matrix (Figure 4b,c). Magnified images of the microstructure show that the reinforcements (Ti1400 and TiC particles) are tightly bonded to the substrate without cracks, pores, or other defects. This good interfacial bonding is due to the tensile stress generated in the RD direction and the compressive stress in the ND direction during the hot rolling process, which aids in the tight bonding between the reinforcement and the matrix. The in situ generated TiC presented as round particles with an average size of about 1.5 µm (Figure 4d,e). Inside the TC4 matrix and at the border between the matrix and the Ti1400 region, the TiC particles were evenly dispersed. The Ti400 region included no TiC particles (Figure 4f).

### 3.2. Mechanical Properties of (TiC + Ti1400)/TC4 Composites

The mechanical properties of TMC-0, TMC-3, and TMC-20 composites are presented in Figure 5 and Table 2. All the samples showed high yield strength and tensile strength (Figure 5a). The yield strength and tensile strength of the TMC-0 samples reached 1361 MPa and 1477 MPa, respectively, with an elongation of about 8.4%. The TMC-3 sample performed better, with yield strength and tensile strength of 1358 MPa and 1524 MPa, respectively, and an elongation of about 9%. The TMC-20 sample had a yield strength and tensile strength of 1267 MPa and 1477 MPa, respectively, and an elongation of about 5.7%. Figure 5b shows the average values of the mechanical properties, which can indicate that the specimens are more repeatable and the properties are more stable. With the addition of Ti1400, the yield strength of the TMC-3 sample was not much different from that of the TMC-0 sample, but the tensile strength was increased by about 47 MPa, and the elongation was also improved. This indicates that a small addition of Ti1400 can increase the strength of the composite while minimizing the loss of elongation. The elongation of the TMC-20 sample was 32% lower compared to the TMC-0 sample, and the yield strength was 7% lower than that of TMC-0. At higher Ti1400 content, the composites showed severe plasticity loss.

The fracture surfaces of the samples are shown in Figure 6a, where the TMC-3 sample exhibits significant necking and the TMC-20 sample shows no significant deformation. Observing the fracture surface, the proportion of the TMC-3 fiber area increased significantly (Figure 6b). In contrast, the macroscopic fracture surface of TMC-20 is relatively flat and exhibits granular features, predominantly radial zones with rough radial ridges (Figure 6c). These features indicate the nature of brittle fracture. In Figure 6d–f, TMC-3 forms many equiaxed dimples under tensile stress. These dimples are distributed with second-phase particles, and fine TiC particles are also observed in the dimples, which indicates that the bonding between the TiC particles and the matrix is still tight even during the fracture process, thus providing reformation resistance and load transfer capability. Compared to the fracture surface of TMC-3, some areas of TMC-20 showed larger and deeper dimples as well as distinct tear ridges, which indicate that the sample has excellent toughness. However, the dimples on the tear ridges show signs of stretching, suggesting an uneven distribution of stresses within the sample during stretching (Figure 6g–i). Similar conclusions were drawn for the titanium matrix composites prepared by Krystian Zyguła et al., and the diffusion behavior and possible precipitation capacity of the TiC and TiB phases during hot pressing were investigated in detail [47].

The fracture mechanism for TMC-3 is microporous aggregation fracture, characterized by uniformly distributed dimples and fewer tear ridges. This fracture mechanism is typical of ductile fracture and suggests that the material has excellent plasticity, which is consistent with the excellent elongation observed in the tensile tests. Further macroscopic observation of the TMC-20 fracture reveals an icing sugar pattern that differs from the conventional icing sugar pattern fracture, primarily due to the finer and longer grains along the RD (Figure 6g). The stress concentration phenomenon depicted in Figure 6i diminishes the overall cohesive strength of the material, potentially leading to premature failure under tensile stress. This could result in localized deformation and premature fracture, potentially accounting for the reduced elongation of the TMC-20 samples.

The schematic diagram of fracture is shown in Figure 7. The fine grains effectively enhance the material’s strength by hindering the movement of dislocations. After rolling, the continuous network of TiC reinforcement becomes dispersed and uniformly distributed in the matrix. These uniformly distributed TiC particles are known for their greater resistance to deformation and load-bearing capacity. Due to the deflection and hindrance of fine grains and TiC particles, the crack propagation path exhibits a “zigzag” pattern, thereby significantly improving the material’s toughness. In TMC-3, when the Ti1400 region exists in a small and uniformly dispersed form, it does not hinder the plastic deformation of the composites. The β-phase formed by the Ti1400 region acts as a soft phase, promoting dislocation slip by coordinating deformation, which helps maintain or enhance the ductility of the composites. However, in TMC-20, an excess of Ti1400 may lead to aggregation in the matrix, and the longer continuous interface of Ti1400 provides a direct path for crack propagation, reducing the possibility of crack deflection. This structural change may make the material more inclined to brittle fracture under tensile stress, as the crack can more easily expand along the weaker interface. The fracture mechanism of TMC-20 is characterized by trans granular fracture in the matrix, while in smaller Ti1400 regions, the crack continues to deflect. In Ti1400-rich areas, the crack may rapidly propagate along the interface between the matrix and Ti1400, ultimately leading to fracture. Therefore, the fracture mode of TMC-20 should be classified as a mixed fracture mode of trans granular ductile fracture and intergranular brittle fracture.

### 3.3. Strengthening Mechanism of (TiC + Ti1400)/TC4 Composites

The strength and ductility of the (Ti1400 + TiC)/TC4 composite were related to the strength and content of Ti1400. In the composites, an increase in the strength of the Ti1400 region would be beneficial for enhancing the overall strength of the composites. The ductility of the Ti1400 region was much better than that of TiC, enabling it to coordinate deformation through plastic deformation under higher stress concentrations, thereby improving the ductility of the composites. If the strength of the Ti1400 region was relatively low, the strength of the composites would have decreased as the content of Ti1400 increased.

This study tested the Vickers hardness of the TC4 matrix, the Ti1400 region, and their interface in the composites. Figure 8a was a schematic diagram showing the locations where hardness was tested. The Vickers hardness values for the interior of the Ti1400 region, its interface with the TC4 matrix, and the TC4 matrix were 349 HV, 360 HV, and 431 HV, respectively, as shown in Figure 8b. The Vickers hardness of the TC4 matrix was significantly higher than that of the Ti1400 region, with a difference of nearly 100 HV. The strength of the Ti1400 region was lower than that of the TC4 matrix because Ti1400 was a near-β titanium alloy with a low Mo equivalent. After rolling at 1173 K and cooling, the Ti1400 region retained more β-phase, which had lower strength in the solid solution state, leading to a reduction in the strength of the Ti1400 region. Therefore, the presence of 3 wt.% Ti1400 did not significantly affect the strength of the composites, but when its content increased to 20 wt.%, it reduced the strength of the composites. Subsequent work will involve aging treatment of the composites, which will significantly enhance the strength of the Ti1400 region and thereby contribute to improving the strength of the composites.

In Figure 9a,b, the Ti1400 region in the composites was mainly composed of the β-phase and was distributed along the rolling direction. Due to the absence of further annealing treatment, the composites were still dominated by deformed structures, with only a small portion of recrystallized structures, as shown in Figure 9c. Therefore, the composites still exhibited high residual stress, as illustrated in Figure 9d. Dislocations generated by rolling deformation accumulated near TiC and at α/β interfaces, and these dislocations were not fully recovered after hot rolling. The remaining dislocations were mainly stored near TiC and at α/β interfaces, with fewer within the β-phase. This provided the possibility for subsequent coordinated deformation of the β-phase. When a small amount of Ti1400 was added, the Ti1400 regions were dispersed in the matrix. These smaller regions and their interfaces with the matrix exerted weaker hindrance to dislocations, thus not significantly reducing the strength of the composites. Due to the small number and uniform dispersion of Ti1400 regions, they did not hinder plastic deformation of the composites but instead promoted dislocation slip through coordinated deformation, helping to maintain or improve the ductility of the composites.

Additionally, due to the relatively low stability of the β-phase in the solid solution state, under the influence of surrounding stress concentrations, coordinated deformation may lead to martensitic transformation (such as the TRIP effect). During stress loading, stress-induced martensitic transformation occurred, and the formed lath-shaped martensite cut the β-phase grains. This resulted in the dynamic Hall–Petch effect [48,49,50,51], which caused the Ti1400 region to exhibit significant work hardening and higher strength, thereby enhancing the strength of the composites and also contributing to a certain degree of work-hardening capability (for TMC-3:TS-YS = 166 MPa).

Conversely, when a large amount of Ti1400 was added, the number of Ti1400 regions in the composites increased significantly, potentially leading to the merging of Ti1400 regions locally and resulting in uneven distribution. Although this can increase the work-hardening capability of the composites (for TMC-20:TS-YS = 210 MPa), the longer continuous interfaces provided more sources for crack initiation and paths for crack propagation, reducing the toughness of the material and making the composites more prone to fracture. Therefore, the content of Ti1400 in the composites should not be excessive. Its uniform distribution within the composites can not only increase strength but also improve ductility and exhibit good work-hardening capability.

## 4. Conclusions

This study utilized low-energy ball milling and spark plasma sintering techniques to fabricate in situ synthesized (TiC + Ti400)/TC4 composites, which were then subjected to hot rolling. The hot-rolled composites with varying Ti1400 contents were characterized and tested using OM, SEM, EBSD, Vickers hardness, and room temperature tensile tests. The strengthening mechanisms of the composites were analyzed. The main conclusions are as follows:(1)The Ti1400 alloy bonds well with the matrix, forming uniformly distributed Ti1400 regions within the matrix. As the content of Ti1400 increases, these regions enlarge and begin to coalesce locally. The in situ reaction-produced TiC particles are discontinuously distributed around the TiC-lean regions, creating a three-dimensional network structure. The size of the network units is similar to the diameter of the original spherical TC4 powders.(2)The addition of 3 wt.% Ti1400 alloy can increase the strength of the TiC/TC4 composites to 1524 MPa without reducing the elongation, which remains at a high level of 9%. Moreover, it exhibits enhanced work-hardening performance. However, the addition of 20 wt.% Ti1400 does not improve the tensile strength of the (TiC + Ti1400)/TC4 composites and significantly decreases the elongation of the composites.(3)A small amount of Ti1400 content does not reduce the strength of the composites. The fine α grains and TiC enhance strength and toughness by impeding crack growth, while the β-phase in Ti1400 allows for deformation and stress redistribution, providing higher work hardening and preventing premature failure.

## Figures and Tables

**Figure 1 materials-18-00051-f001:**
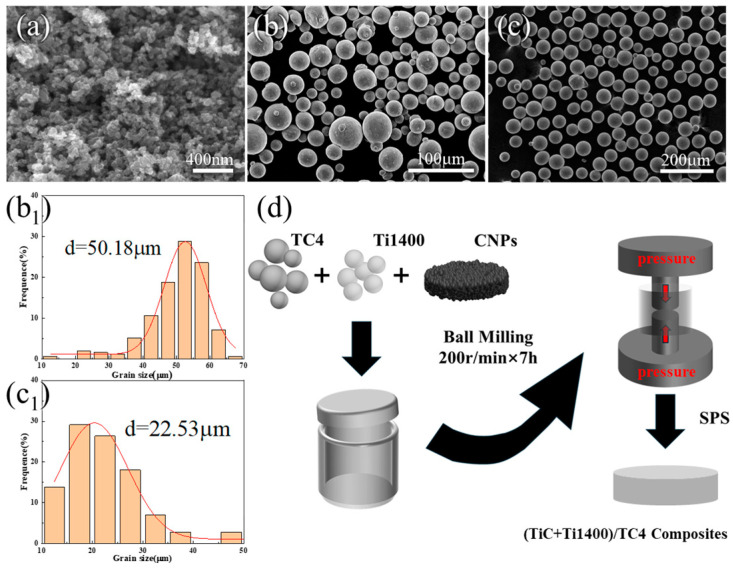
The SEM images of (**a**) CNP powders, (**b**) TC4 powders, (**c**) Ti1400 powders, and (**d**) the preparation process of (TiC + Ti1400)/TC4 composites; (**b_1_**) histogram of TC4 powders’ size distribution; (**c_1_**) histogram of Ti1400 powders’ size distribution.

**Figure 2 materials-18-00051-f002:**
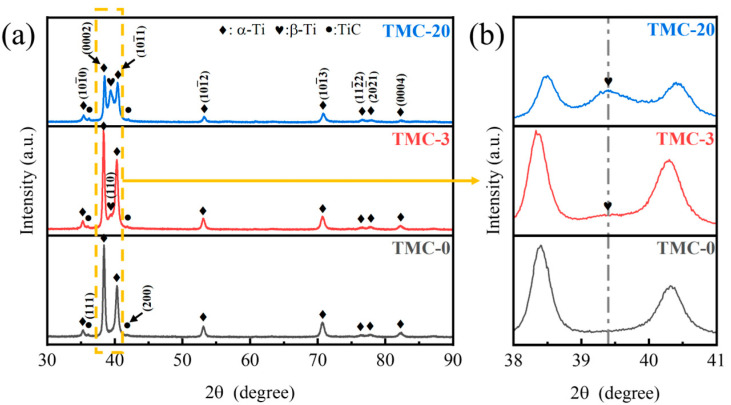
(**a**) XRD spectra of (TiC + Ti1400)/TC4 composites, (**b**) the enlarged view of the yellow box in (**a**).

**Figure 3 materials-18-00051-f003:**
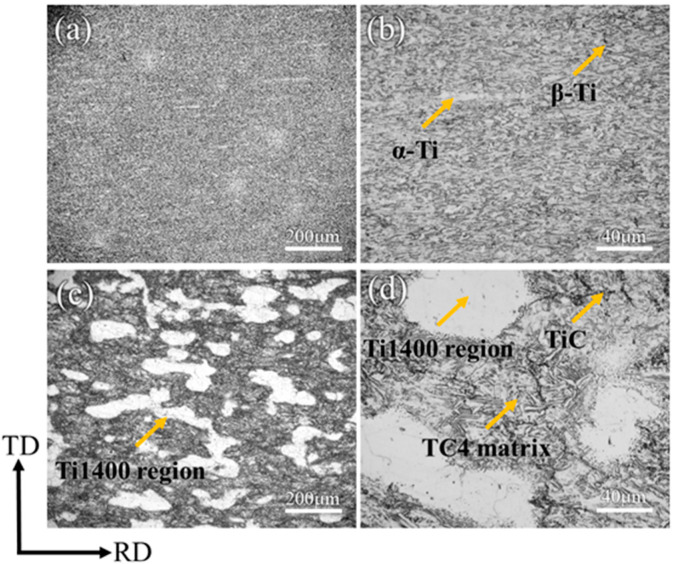
OM images of (**a**,**b**) TMC-0 and (**c**,**d**) TMC-20.

**Figure 4 materials-18-00051-f004:**
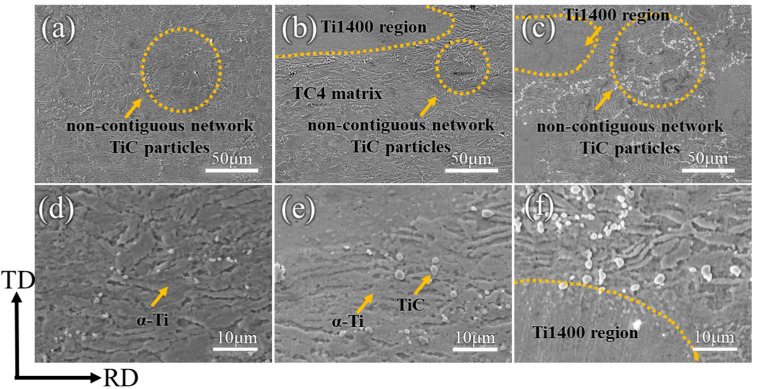
SEM images of (**a**,**d**) TMC-0, (**b**,**e**) TMC-3, and (**c**,**f**) TMC-20.

**Figure 5 materials-18-00051-f005:**
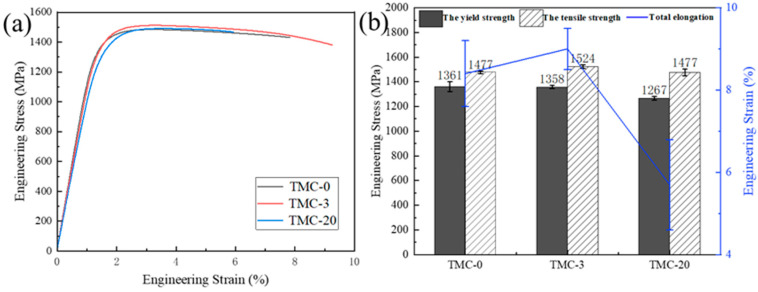
Mechanical properties of (TiC + Ti1400)/TC4 composites. (**a**) Engineering stress–strain curves; (**b**) mean and standard deviation of yield strength, tensile strength, and elongation of composites.

**Figure 6 materials-18-00051-f006:**
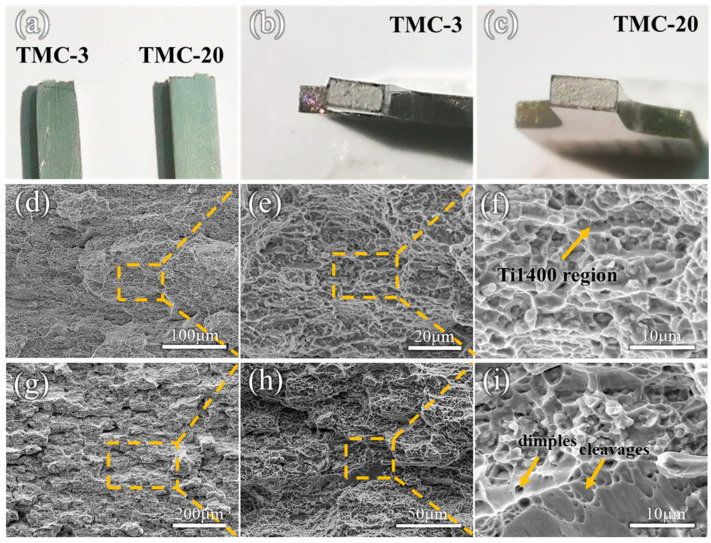
(**a**–**c**) The macroscopic fracture morphology of the sample; SEM images of the tensile fracture of (**d**–**f**) TMC-3 and (**g**–**i**) TMC-20.

**Figure 7 materials-18-00051-f007:**
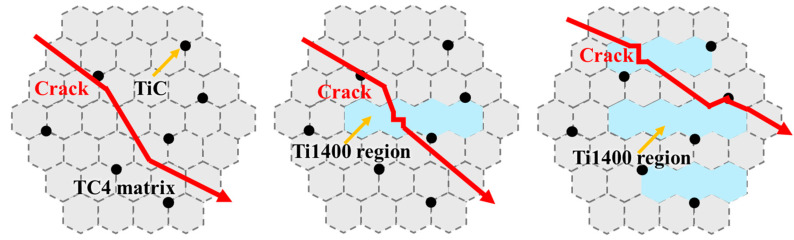
The schematic diagram of the fracture mode for the (TiC + Ti1400)/TC4 composites.

**Figure 8 materials-18-00051-f008:**
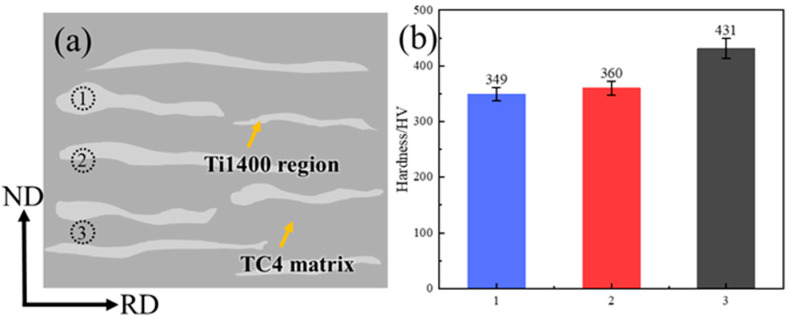
Vickers hardness plots of TMC-20 samples at different locations. (**a**) Schematic diagram of the sampling location; (**b**) Statistical plot of Vickers hardness, numbers 1–3 correspond to the Vickers hardness test positions in Figure (a).

**Figure 9 materials-18-00051-f009:**
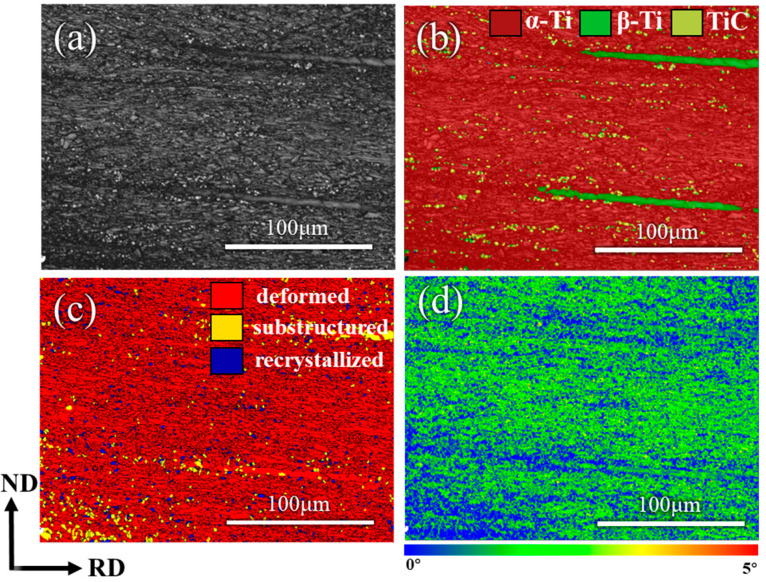
EBSD images of the TMC-3 sample. (**a**) Band contrast image; (**b**) phase image; (**c**) recrystallization image; (**d**) kernel average misorientation image.

**Table 1 materials-18-00051-t001:** The composition and content of the TC4 and Ti1400 powders.

Element (wt. %)	Al	V	Fe	Cr	Mo	Sn	Zr	Ti
TC4	6.07	3.93	0.18	/	/	/	/	leftover
Ti1400	4.12	4.62	0.18	5.55	4.16	<0.01	<0.01	leftover

**Table 2 materials-18-00051-t002:** Mechanical properties of (TiC + Ti1400)/TC4 composites.

Samples	Ti1400 Mass Fractions/wt.%	The Yield Strength (YS)/MPa	The Tensile Strength (TS)/MPa	Total Elongation (tEl)/%
TMC-0	0	1361 ± 39	1477 ± 9	8.4 ± 0.8
TMC-3	3	1358 ± 13	1524 ± 15	9.0 ± 0.5
TMC-20	20	1267 ± 15	1477 ± 26	5.7 ± 1.1

## Data Availability

The original contributions presented in the study are included in the article, further inquiries can be directed to the corresponding authors.

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
