# Peer review of "Microstructure and Mechanical Properties of Rolled (TiC + Ti1400)/TC4 Composites"

_materials, 2024, doi:10.3390/ma18010051_

Round 1
Reviewer 1 Report
Comments and Suggestions for Authors
In the manuscript, the authors obtained Ti1400 alloy and in situ synthesized TiC reinforced (TiC+Ti1400)/TC4 composite using low energy ball milling and spark plasma sintering followed by hot rolling to obtain titanium matrix composites. Composites with different Ti1400 contents were characterized and tested using OM, SEM, EBSD, Vickers hardness and room temperature tensile tests. The authors obtained interesting and meaningful results. The work is written at a high scientific level, all the results are analyzed and discussed in detail by the authors and significant conclusions are drawn.
I have the following comments on the manuscript:
1. Figure 1. Axis captions on insets with histograms of powder size distribution are not visible. I think it would be better to make the histograms a separate figure.
2. Materials preparation. “The hot rolling temperature was 120 1173 K with a total deformation of 75%.” Here the authors must describe the rolling regime in more detail: how many passes of reduction were used, the final thickness of the sheet.
3. Figure 9d. Authors must include a color legend for the kernel average misorientation map.
4. Figure 9c. Authors must identify colors for recrystallized non-crystallized and sub-grained regions.
5. Figure 5b. Authors must indicate which of the columns corresponds to the yield strength and which to the tensile strength.
6. Figure 8. Confidence intervals for hardness values must be added.
Author Response
Comments 1: Figure 1. Axis captions on insets with histograms of powder size distribution are not visible. I think it would be better to make the histograms a separate figure.
Response 1: Thank you very much for the reminder. Based on your comment, we have corrected figure 1. The changes are in line 113 of the manuscript.
Comments 2: Materials preparation. “The hot rolling temperature was 120 1173 K with a total deformation of 75%.” Here the authors must describe the rolling regime in more detail: how many passes of reduction were used, the final thickness of the sheet.
Response 2: Thank you for your rigorous questions. We have added details about the rolling section. The size of the sintered samples was Φ 50 × 12 mm. The cylindrical samples of 8 mm in thickness were cut from the sintered samples for rolling. Before rolling, the samples were kept warm in a muffle furnace at 1173 K for 20 min. The samples were rolled from 8 mm to 2 mm in 9 passes by a two-high mill with a model of Φ 200 × 300, and the deformation was 75%. The samples for tensile testing had been cut in the rolling direction. The relevant additions are located on line 130 in the manuscript.
Comments 3: Figure 9d. Authors must include a color legend for the kernel average misorientation map.
Response 3: Thank you for your sincere suggestion, we have added the color legend on Figure 9d. Modified image at line 316 in the manuscript.
Comments 4: Figure 9c. Authors must identify colors for recrystallized non-crystallized and sub-grained regions.
Response 4: Thank you very much for your attention, we have corrected Figure 9c. Modified image at line 316 in the manuscript.
Comments 5: Figure 5b. Authors must indicate which of the columns corresponds to the yield strength and which to the tensile strength.
Response 5: Thank you very much for the reminder! We have corrected the image, and we apologize for any inconvenience caused to your reading. Modified image at line 221 in the manuscript. Thanks again.
Comments 6: Figure 8. Confidence intervals for hardness values must be added.
Response 6: Thanks for your careful checks. We have added the confidence intervals for the hardness values in Figure 8. Modified image at line 299 in the manuscript.

Reviewer 2 Report
Comments and Suggestions for Authors
Dear Authors,
Thank you for the opportunity to review the paper entitled “Microstructure and mechanical properties of rolled (TiC+Ti1400)/TC4 composites”. Below you will find a few remarks regarding your work.
The aim of the study is not clearly specified. The last paragraph of the introduction presents a summary of what the Authors plan to do, but do not indicate why they are going to do so. What is the purpose of the study?
Table 1 – was the chemical composition of powders measured by the Authors or provided by the manufacturer? The method of analysis should be mentioned in the text. Regarding TC4 – does the “/” mean that it is 0 or that it was not measured?
Line 120 – why prepare the cylindrical samples if it is going to be subjected to hot rolling? wouldn’t it be easier to prepare a flat sample?
Line 121 – was the 75% deformation applied in one pass or multiple passes?
Were the samples for tensile testing cut along the rolling direction? When analyzing sheet materials, the anisotropy of properties should be taken into account. The samples should be evaluated along the rolling direction (0°), perpendicular to the rolling direction (90°) and at the 45° to the rolling direction.
Fig. 3 – I believe using black color font would be better, since it is extremely difficult to read the white one which is currently used.
Line 175 – from magnified image (fig 4d) I would not be so sure that the β-phase is elongated along the rolling direction. It is elongated mostly in the rolling direction, but there are some β-phases that are elongated randomly.
Fig 5 b – I believe gray is YS and red is UTS, but it should be specified somewhere.
Table 2 – if the Authors specified the nomenclature of the samples as TMC-0, TMC-3 and TMC-20 then it should be used here as well instead of “serial number 1, 2, 3.”
Overall, the results and discussion section lacks comparison of the obtained results with papers published by others. The whole section has only one sentence with references to other papers. The topic of titanium alloys and titanium matrix composites is widely studied, and it should be easy for the Authors to find some works to which their results can be compared to.
References
There are 42 references and all of them are quite recent, mostly published in the past 5 years. There are 4 self-citations, which gives less than 10% - in my opinion it is acceptable. Overall, good choice of cited papers.,
Author Response
Comments 1: The aim of the study is not clearly specified. The last paragraph of the introduction presents a summary of what the Authors plan to do, but do not indicate why they are going to do so. What is the purpose of the study?
Response 1: Thank you very much for the reminder. Previously, He et al. prepared (Ti1400+TiC)/TC4 by spark plasma sintering and the properties were tensile strength of 1178 MPa and elongation of 12.8%. However, for TMCs, strength over 1500 MPa and elongation over 5% are still a challenge. Thermo-mechanical processing, such as hot rolling, is a common method to modulate the microstructure and mechanical properties of titanium alloys and TMCs. The strengthening mechanisms it brings, such as grain refinement and dislocation strengthening, will be an effective means to achieve the extraordinary mechanical properties of TMCs. Therefore, in this paper, (Ti1400+TiC)/TC4 was prepared by the SPS method, and in order to avoid the cracking due to the poor plasticity of TiC and the severe oxidation of the titanium alloy matrix during the hot rolling process, multiple passes were carried out below the temperature of the phase transition point of TC4 (1173 K). Relevant content has been added to line 86 in the manuscript.
Comments 2: Table 1 – was the chemical composition of powders measured by the Authors or provided by the manufacturer? The method of analysis should be mentioned in the text. Regarding TC4 – does the “/” mean that it is 0 or that it was not measured?
Response 2: Thanks for your helpful comment. Based on your comments, relevant content has been added to line 105 in the manuscript. The manufacturer provided the chemical elements of the TC4 original powder. The chemical elements of Ti1400 powder were measured by the authors by inductively coupled plasma atomic emission spectrometry. In Table 1, the “/” means that the element was not measured.
Comments 3: Line 120 – why prepare the cylindrical samples if it is going to be subjected to hot rolling? wouldn’t it be easier to prepare a flat sample?
Response 3: Thank you for your rigorous question. In order to ensure that the SPS sintered samples have high densities, a pressure of 40 MPa needs to be applied during the sintering process. Flat samples have a larger area, which requires a higher pressure on the equipment. The sample sizes in this paper were determined after considering the rolling deflection and the maximum pressure of our existing SPS sintering equipment. Thank you for your question.
Comments 4: Line 121 – was the 75% deformation applied in one pass or multiple passes? Were the samples for tensile testing cut along the rolling direction? When analyzing sheet materials, the anisotropy of properties should be taken into account. The samples should be evaluated along the rolling direction (0°), perpendicular to the rolling direction (90°) and at the 45° to the rolling direction.
Response 4: Thank you for your rigorous questions. We have added details about the rolling section. The size of the sintered samples was Φ50 × 12 mm. The cylindrical samples of 8 mm in thickness were cut from the sintered samples for rolling. Before rolling, the samples were kept warm in a muffle furnace at 1173K for 20 min. The samples were rolled from 8 mm to 2 mm in 9 passes by a two-high mill with a model of Φ 200 × 300, and the deformation was 75%. The samples for tensile testing had been cut in the rolling direction. The relevant additions are located on line 130 in the manuscript. Other editors have mentioned this, so the additions are marked yellow in the manuscript.
Comments 5: Fig. 3 – I believe using black color font would be better, since it is extremely difficult to read the white one which is currently used.
Response 5: Thank you very much for the reminder! We have modified the image with the relevant problem, and we apologize for any inconvenience caused to your reading. Thanks again.
Comments 6: Line 175 – from magnified image (fig 4d) I would not be so sure that the β-phase is elongated along the rolling direction. It is elongated mostly in the rolling direction, but there are some β-phases that are elongated randomly.
Response 6: Thank you for your rigorous question. This part of the description in the manuscript is not accurate enough. Figure 4 is observed from the ND direction, and part of the β-phase after rolling is elongated along the RD direction, and part of it is elongated along the TD direction. Relevant content has been modified at line 186 in the manuscript.
Comments 7: Fig 5 b – I believe gray is YS and red is UTS, but it should be specified somewhere.
Response 7: Thank you for your sincere suggestion, we have added the color legend on Figure 5b. Modified image at line 221 in the manuscript. Other editors have mentioned this, so the additions are marked yellow in the manuscript.
Comments 8: Table 2 – if the Authors specified the nomenclature of the samples as TMC-0, TMC-3 and TMC-20 then it should be used here as well instead of “serial number 1, 2, 3.”
Response 8: Thanks for your helpful comment. Based on your suggestion, Table 2 has been modified. Corrected table 2 is located on line 224 of the manuscript. Thanks for your correction.
Comments 9: Overall, the results and discussion section lacks comparison of the obtained results with papers published by others. The whole section has only one sentence with references to other papers. The topic of titanium alloys and titanium matrix composites is widely studied, and it should be easy for the Authors to find some works to which their results can be compared to.
Response 9: Thank you for your valuable comment! As suggested by the reviewers, we added [1-9] relevant literature to support our ideas. These papers were added to the Introduction and Mechanical properties of (TiC+Ti1400)/TC4 composites sections of the manuscript. Thank you for your suggestions.
References
- Chen, Y.; Zhang, H.; Wang, B.; Huang, J.; Zhou, M.; Wang, L.; Xi, Y.; Jia, H.; Xu, S.; Liu, H.; Wen, L.; Xiao, X.; Liu, R.; Ji, J., A Review of Research on Improving Wear Resistance of Titanium Alloys. Coatings 2024, 14, (7).
- Golyshev, A.; Malikov, A.; Vitoshkin, I., The Use of Boron Fibers and Particles for Creating Functionally Graded Material Based on Ti64 Using the Laser Additive Manufacturing Method. Crystals 2023, 13, (7).
- Shetty, R.; Hegde, A.; Shetty Sv, U. K.; Nayak, R.; Naik, N.; Nayak, M., Processing and Mechanical Characterisation of Titanium Metal Matrix Composites: A Literature Review. Journal of Composites Science 2022, 6, (12).
- Yan, H.; Wang, L.; Wang, X.; Jiang, B.; Liu, H.; Wang, B.; Luo, L.; Su, Y.; Guo, J.; Fu, H., Effects of Melt Hydrogenation on the Microstructure Evolution and Hot Deformation Behavior of TiBw/Ti-6Al-4V Composites. Materials (Basel) 2023, 16, (6).
- Zygula, K.; Mrotzek, T.; Lypchanskyi, O.; Zientara, D.; Gude, M.; Prahl, U.; Wojtaszek, M., Microstructure and Mechanical Properties of In Situ Synthesized Metastable beta Titanium Alloy Composite from Low-Cost Elemental Powders. Materials (Basel) 2023, 16, (23).
- Mihai, S.; Baciu, F.; Radu, R.; Chioibasu, D.; Popescu, A. C., In Situ Fabrication of TiC/Ti-Matrix Composites by Laser Directed Energy Deposition. Materials (Basel) 2024, 17, (17).
- Bernard, G.; Pejchal, V.; Sereda, O.; Loge, R. E., Tensile Properties of Ex-Situ Ti-TiC Metal Matrix Composites Manufactured by Laser Powder Bed Fusion. Materials (Basel) 2024, 17, (22).
- Ozerov, M.; Stepanov, N.; Sokolovsky, V.; Astakhov, I.; Klimova, M.; Galtsev, A.; Huang, L.; Zherebtsov, S., Deformation Behavior and Microstructure Evolution of a TiB-Reinforced Ti-6.5Al-2Zr-1Mo-1V Matrix Composite. Metals 2023, 13, (11).
- Li, Z.; Xing, S.; Wu, S.; Hou, J.; Wu, S., A Review on the Interface Structure and Control Between Graphene Nanoplatelets (GNPs) and Ti Matrix of GNPs/Ti Matrix Composites. Metals 2024, 14, (12).

Round 2
Reviewer 1 Report
Comments and Suggestions for Authors
The authors significantly corrected and improved the manuscript. I think the manuscript can be accepted for publication.
Reviewer 2 Report
Comments and Suggestions for Authors
Dear Authors,
thank you for providing the revised version. You have answered all of my questions.